# Generalized Deep Learning EEG Models for Cross-Participant and Cross-Task Detection of the Vigilance Decrement in Sustained Attention Tasks

**DOI:** 10.3390/s21165617

**Published:** 2021-08-20

**Authors:** Alexander Kamrud, Brett Borghetti, Christine Schubert Kabban, Michael Miller

**Affiliations:** Department of Electrical and Computer Engineering, Air Force Institute of Technology, Wright-Patterson Air Force Base, OH 45433, USA; brett.borghetti@afit.edu (B.B.); christine.schubert@afit.edu (C.S.K.); michael.miller@afit.edu (M.M.)

**Keywords:** EEG, deep learning, vigilance decrement, sustained attention, mental fatigue, cross-participant, cross-task, task-generic

## Abstract

Tasks which require sustained attention over a lengthy period of time have been a focal point of cognitive fatigue research for decades, with these tasks including air traffic control, watchkeeping, baggage inspection, and many others. Recent research into physiological markers of mental fatigue indicate that markers exist which extend across all individuals and all types of vigilance tasks. This suggests that it would be possible to build an EEG model which detects these markers and the subsequent vigilance decrement in any task (i.e., a task-generic model) and in any person (i.e., a cross-participant model). However, thus far, no task-generic EEG cross-participant model has been built or tested. In this research, we explored creation and application of a task-generic EEG cross-participant model for detection of the vigilance decrement in an unseen task and unseen individuals. We utilized three different models to investigate this capability: a multi-layer perceptron neural network (MLPNN) which employed spectral features extracted from the five traditional EEG frequency bands, a temporal convolutional network (TCN), and a TCN autoencoder (TCN-AE), with these two TCN models being time-domain based, i.e., using raw EEG time-series voltage values. The MLPNN and TCN models both achieved accuracy greater than random chance (50%), with the MLPNN performing best with a 7-fold CV balanced accuracy of 64% (95% CI: 0.59, 0.69) and validation accuracies greater than random chance for 9 of the 14 participants. This finding demonstrates that it is possible to classify a vigilance decrement using EEG, even with EEG from an unseen individual and unseen task.

## 1. Introduction

Mental fatigue is a significant contributor to a decline in performance for sustained attention type tasks [1,2], also known as vigilance tasks. Vigilance tasks require operators to remain focused and alert to stimulus during a task [3], and in the control and surveillance of today’s automated systems, vigilance typically suffers either due to the low level of workload and stimulus associated with the task [4] or due to the mental demands vigilance requires over a lengthy task [5].

A decline in performance during these vigilance tasks is called a vigilance decrement, and it is defined as a decrease in probability of detecting rare but significant events within vigilance tasks [6]. Some form of mental fatigue is typically associated with a vigilance decrement, and this mental fatigue has been linked to increased human error rate [7,8,9]. If this mental fatigue could be detected using artificial intelligence (AI), then systems could be developed to regulate mental fatigue by varying levels of stimulus to aid in sustained attention [10,11] or by providing recovery time [12].

Mental fatigue has also been linked to specific changes in physiological measures, such as specific increases and decreases in magnitude for the average spectral power of different frequency bands for electroencephalography (EEG) signals [13,14]. Recent machine learning research has utilized EEG signals to classify mental fatigue in specific tasks such as driving [15]; however, a task-generic model which can accurately classify either mental fatigue or a vigilance decrement through EEG signals has not yet been generated. The EEG markers of mental fatigue during vigilance tasks are consistent across both participants and different types of tasks, and mental fatigue is typically always associated with a vigilance decrement in vigilance tasks [16], thus, a model could be built which is capable of performing classification of a vigilance decrement in any vigilance task, through detection of mental fatigue in EEG signals, in any individual’s EEG (i.e., a cross-participant model).

Recently, Yin et al., pursued the goal of building a task-generic cross-participant mental fatigue detector using extreme learning machines (ELMs) [17]. Two tasks were used which had the participants replicate the role of an aircraft’s automated cabin air management system. Eight participants performed Task 1, and six different participants performed Task 2. Each task varied parameters within the task to create “low” and “high” mental fatigue conditions, with these conditions then corresponding to labeled trials of their respective condition. Models were then built from the EEG data for each task and each condition using entropy features and spectral features (average power of the theta, alpha, beta, and gamma bands) as input features. The models were then tested upon the participant data of the opposite task, with classification accuracies ranging from 65% to 69%. An issue with relating this to vigilance decrement detection is that the tasks simply varied parameters within the task to create “low” and “high” mental fatigue conditions. These conditions were then used as the labels to train the classifier. This means the classifier was trained to identify EEG signals which correspond to these “low” and “high” mental fatigue conditions and not actual vigilance decrements. For proper identification of a vigilance decrement, instead an objective measure of the participant’s performance which is associated with the vigilance decrement (such as accuracy and/or response time) would need to be recorded and used to generate the labels of vigilance decrement vs. no decrement for the machine learning classifier. Another issue is that it is unclear if the two stated tasks are analogous to two separate tasks in the real-world, such as the difference between driving and monitoring closed-circuit security cameras, as both tasks used in the experiment had participants performing the same role of the aircraft automated cabin air management system, with only certain parameters and conditions being varied between the two tasks. This suggests that their results are applicable to a varied version of the same type of task that is not truly task-generic.

In this research, we build three different cross-participant models which use EEG signals to perform task-generic classification of the vigilance decrement on any individual. Two of the models are time-domain based, meaning they use the raw EEG time-series voltage values as their data, and the third model is frequency-domain based, using spectral features extracted from the average power of the five clinical EEG frequency bands. The data is comprised of two EEG datasets, with each dataset containing different participants and each dataset containing different vigilance tasks (three different tasks in total). These datasets were collected by the 711th Human Performance Wing (HPW) in partnership with the University of Dayton through two different experiments for the purpose of studying event related potentials (ERPs) during a vigilance decrement across various vigilance tasks [18,19]. Models are trained on data from two of the vigilance tasks and only a subset of the participants and then tested using data from a separate vigilance task that the model has not seen, as well as participants that the model has not seen, which is crucial in order to avoid overestimated test accuracies in cross-participant EEG models [20].

The significant contribution of this research is a model which is capable of detecting a vigilance decrement in unseen participants in an unseen task, as evidenced by the best performing model with 7-fold CV accuracy significantly greater than random chance at 64% (95% CI: 0.59, 0.69). This finding is novel as the cross-participant model was tested with a separate task that the model had not seen, meaning the vigilance decrement was classified in unseen participants in an unseen task, and thus far, a task-generic model which is capable of vigilance decrement classification in an unseen task has not yet been created. Previous work by Yin et al. in building a task-generic model did not utilize a different type of task in order to validate their model and instead only varied parameters within a single task of operating an aircraft’s cabin air management system in order to create two tasks. Our research in contrast utilized three different types of tasks (the air traffic controller task, the line task, and the 3-stimulus oddball task), each of which are well established in the literature as different kinds of vigilance-type tasks.

This paper has the following structure. First, in Section 2, background is provided for the vigilance decrement and how it is linked to EEG. Next, in Section 3, we provide our methodology, first providing details on the datasets collected and the tasks used within those datasets, followed by details for the training and testing of all three models. Then, in Section 4, results are presented for all three models. Finally, in Section 5, results are compared and discussed, with conclusions and future work following in Section 6.

## 2. Related Work

Decision making and how it deteriorates in stressful work environments has been extensively studied since the late 1800s [6]. One of the main phenomena studied has been the concept of vigilance, which is the quality or state of being wakeful or alert. Tasks which require vigilance fall under a taxonomy developed by Parasuraman and Davies [21], with the taxonomy classifying tasks into different categories based on specific information-processing transactions within the tasks themselves, such as signal discrimination (successive or simultaneous), task complexity, event rate, and sensory modality. For signal discrimination, simultaneous tasks are ones in which the critical stimulus and non-critical stimulus are both present at the same time for participants to use for comparison. Successive tasks, however, do not provide these stimulus to the participant at the same time, and therefore, it requires the participant to hold the non-critical stimulus in memory.

### 2.1. Vigilance Decrement

Extensive research over the decades on vigilance and the vigilance decrement has found that the behavioral cause of the decrement is due to performing attention-demanding tasks over an extended period of time, ranging from tens of minutes to hours, depending on the task and its cognitive demand [22]. Performing these attention-demanding tasks for extended periods of time results in mental fatigue and/or a decrease in sustained attention [23], with mental fatigue being defined as a gradual and cumulative phenomenon that is increased in magnitude by time spent on a tedious but mentally demanding task [24].

Numerous factors have also been found to affect the magnitude and timing of the vigilance decrement [25]. For magnitude, simultaneous stimulus, shorter signals [26,27], task type/source complexity [28], and stimulus event rate [29,30], all result in a greater vigilance decrement. For timing, the vigilance decrement varies depending on the task demands, with the vigilance decrement occurring earlier in more difficult tasks [22], and typically occurring within the first 20–35 min of a task, with half of the decrement occurring in the first 15 min [31].

#### Performance Measurement

To identify in data whether a vigilance decrement has occurred, some measure of task performance through either accuracy, shown in Equation (Equation 1), response time (RT), shown in Equation (Equation 2), or both, is needed. Accuracy and RT are frequently correlated, such that slower responses are more accurate and vice versa, and this is referred to as the speed-accuracy trade-off [32,33].
(1)$$\mathrm{Accuracy}=\frac{\mathrm{hits}\;+\;\mathrm{correct}\;\mathrm{rejections}}{\mathrm{hits}\;+\;\mathrm{false}\;\mathrm{alarms}\;+\;\mathrm{misses}\;+\;\mathrm{correct}\;\mathrm{rejections}}.$$
(2)$$\mathrm{Response}\;\mathrm{Time}=T_{response}-T_{stimulus}.$$

Due to this correlation, it is best to use both accuracy and RT to assess performance, and many different measures have been developed to combine both speed and accuracy into a single measure of performance. For example, there is the Inverse Efficiency Score (IES), which is the ratio of the mean RT and the proportion of correct responses (PC) [34], the Rate-Correct Score (RCS) which is the inverse of the correct RT-based IES [35], the Balanced Integration Score (BIS) which is a combined z-score of RT and accuracy [36], and many others. Recently, research by Mueller et al. examined 12 different measures of accuracy and RT on a vigilance task to determine their sensitivity to the vigilance decrement and found that most single measures which combined accuracy and RT were slight improvements over just accuracy or RT alone [37]. While they found that the Linear Ballistic Accumulator model was the most sensitive and representative measure of the vigilance decrement, they also noted that it was difficult and cumbersome to use and recommended the BIS measure overall.

The BIS measure is designed to give equal weights to both PCs and RTs, hence the name Balanced Integration Score, and is shown below in Equation (Equation 3). First, the PCs and RTs are standardized as shown in Equations (Equation 4) and (Equation 5), with participants *j* and standard deviations *s*, and then once standardized, the standardized RT is subtracted from the standardized PC. This gives the difference in standardized mean correct RTs and PCs. $z_{pc}$ and $z_{rt}$ can be calculated individually for each participant *j*, giving the BIS measurement for only that participant, or across all participants, giving the BIS measurement for the group.
(3)$$BIS_{j}=z_{PC_{j}}-z_{RT_{j}}.$$
(4)$$z_{pc_{j}}=\frac{PC_{j}-\bar{PC}}{s_{pc_{j}}}.$$
(5)$$z_{rt_{j}}=\frac{RT_{j}-\bar{RT}}{s_{rt_{j}}}.$$

When calculating measures such as BIS from data collected during a vigilance task, trials must be binned in some manner for the standardized measures of $z_{pc}$ and $z_{rt}$ to be calculated. A common method is to divide the trials over the duration of the experiment into four time segments (bins) [19,37,38]. Once the number of bins is selected, BIS can then be calculated and compared for each bin to determine whether a vigilance decrement has occurred for the participant; a decreasing BIS indicates a decrement in vigilance. A typical method is to plot the bins on a graph to view the participant’s performance over the course of the task as well as to plot a line of best fit (least squares) to see how their performance trended over the course of the task, with a negative slope indicating a vigilance decrement over the course of the entire task.

### 2.2. EEG

Physiological measurements such as EEG, electrocardiography (ECG), and electrooculography (EOG) have been progressively utilized to better understand the underlying mechanisms of mental fatigue and the vigilance decrement over the past two decades, with EEG receiving significant attention in research for its insight into the status of the brain [16]. EEG signals are a measure of the electrical activity in the brain using electrodes distributed over the scalp, and EEG is often referred to by its different clinical frequency bands, namely delta (2–4 Hz), theta (4–7 Hz), alpha (8–12 Hz), beta (13–29 Hz), and gamma (33–80 Hz). A physiological measurement such as EEG has the advantage of providing a more objective measurement of fatigue than a behavioral measure, as behavioral measures are subjective in nature and left to the experimenter’s or participant’s judgment. EEG studies investigating neural correlates of fatigue have found differing results based on the type of fatigue that the participant is experiencing, with the primary difference being fatigue from sleepiness (sleep fatigue) versus accumulating fatigue from cognitive processes and mental workload (mental fatigue). For example, neural correlates of sleep fatigue have been found to differ based on the task that is being performed. Driver fatigue research found that symptoms associated with sleepiness (e.g., prolonged eye closure) correlated to increases in spectral power for the alpha and beta bands [13], while in pilot fatigue studies, sleepiness was more associated with the opposite effect, with decreases in spectral power for the alpha band [39,40]. Mental fatigue, however, has shown consistent neural correlates of increased spectral power for the alpha band across tasks [16]. This allows for the detection of mental fatigue across tasks and across participants. However, given that both types of fatigue can contribute to changes in performance, such as the vigilance decrement, yet have differing neural correlates, it is important to distinguish sleep fatigue from mental fatigue to reduce confounding variables.

Utilizing these neural correlates of EEG has been useful for both within-participant and cross-participant detection of the vigilance decrement, with all previous research being within-task detection of the vigilance decrement. EEG spectral features have been common features used to detect drowsiness, mental fatigue, and alertness [41,42]. Power spectral density (PSD) in combination with independent component analysis [42], the mean power of the frequency bands and their ratios [41,43,44], power spectral indices of wavelet transforms [45], and full spectrum log power are all spectral features that have been used [46]. Directed connectivity has also been utilized using relative wavelet transform entropy and partial directed coherence to estimate the strength and directionality of information flow between EEG nodes [47,48].

## 3. Methods

### 3.1. Datasets

In this study, two existing EEG datasets are utilized, each collected through experiments conducted previously by the United States Air Force Research Laboratory, 711th Human Performance Wing (HPW), in partnership with the University of Dayton. These experiments were each conducted for the purpose of studying ERPs during a vigilance decrement within various vigilance tasks [18,19]; however, the experiments were conducted separately and did not coincide. All data was de-identified before it was shared with us for our experiments, and because it was de-identified existing data, a Human Research Protection Plan review determined this research to be not involving “human subjects” under US Common Rule (32 CFR 219) on 6 June 2020.

In one experiment, 32 participants (10 men and 22 women, ages ranging from 18 to 36 with a mean of 22.7, with 27 being right-handed) completed three different tasks across a two hour session in the following order: the Hitchcock Air Traffic Controller (ATC) Task [49], the Psychomotor Vigilance Test (PVT) [50], and the 3-Stimulus Oddball Task [51]. The PVT was omitted from our research as the task length was short in duration (<10 min) along with a few amount of trials (<100), making it difficult to segment into bins and quantify with the BIS measure. The Hitchcock ATC task and 3-Stimulus Oddball Task were performed as described in Section 3.2.1 and Section 3.2.2, and trials for each task occurred as follows. The ATC Task included 200 practice trials with feedback provided every 50 trials, then a short break followed by 1600 trials without feedback or breaks. The 3-Stimulus Oddball Task included 20 practice trials, a short break, and 4 blocks of 90 trials each, with performance feedback after each block. Practice trials across both tasks are not utilized in analysis or model training. Some participants had incomplete data, and only the data from the 14 participants with complete datasets were analyzed.

The second experiment consisted of two sessions for each participant, conducted over two separate days, and utilized the line task described in Section 3.2.3. Each day, participants performed 200 practice trials and 4 blocks of 400 experimental trials each, with a short few minute break offered between each block. There were 29 participants; however, only 26 of the participants returned the second day. The data from all 29 participants was utilized in the current study. Participant demographics were not available for this.

Experiment details and EEG/ERP analysis can be found in references [18,19], with summary information provided here. For both datasets, the tasks were presented on an LCD 60 Hz monitor using Psychophysics Toolbox [52] within MATLAB. EEG was recorded using a BioSemi Active II 64 + 2 electrode cap (10–20 system) with the 2 reference electrodes placed over the mastoids (with no additional detail provided), with a sampling rate of 512 Hz. Vertical EOG (VEOG) and Horizontal EOG (HEOG) were also recorded [18,19]. Baseline resting EEG was recorded before starting the experiment and checked for artifacts. Voltage offsets were reduced to less than 40 mV to ensure low impedance, and any high impedance electrodes were re-gelled and re-applied.

### 3.2. Vigilance Decrement Tasks

#### 3.2.1. Hitchcock Air Traffic Controller Task

The Hitchcock ATC Task was designed to test theories surrounding sustained attention, workload, and performance, within a standardized controllable task that is relatively more representative of the real world [53]. Stimulus of a filled red circle and three concentric white circles are continually displayed to the participant. Two white line segments are then displayed over these stimuli in different configurations, as seen in Figure 1. The red circle represents a city, and the white line segments represent aircraft. Participants are instructed to respond (through press of a key on a keyboard) only if the two jets are on a collision course with one another, i.e., the white lines are colinear. If they are, this is a critical event, and a small minority of trials are critical events (3.3%), the rest being non-critical as seen in Figure 1. The stimulus appear every 2 s and only remain on screen for 300 ms.

#### 3.2.2. 3-Stimulus Oddball Task

The 3-Stimulus Oddball Task was designed to assess how individuals discriminate targets, non-target distractors, and standard distractors, in various challenging scenarios [51]. In this task, three different visual stimuli can appear: targets, non-target distractors, and standard distractors. Targets and non-target distractors each appear separately in 10% of trials, and standard distractors appear in the remaining 80% of trials. As seen in Figure 2, the target is a large circle, the standard distractor a small circle, and the non-target distractor a large square. Stimuli are every 2 s with a 75 ms duration. Participants are instructed to respond only to targets by pressing a response key on a keyboard, ignoring non-target distractors and standard distractors.

#### 3.2.3. Line Task

In the line task, participants observe a series of pairs of parallel lines and select whether or not each stimulus is critical. The critical stimuli vary among four conditions for the task, and with critical stimuli comprising 10% of the stimuli. The parallel lines are 0.75 mm in width and variable in length based on trial condition [29]. The first and second conditions are successive-discrimination tasks, meaning the participant has to hold the critical stimulus in memory. In the first condition, the set of lines both being 1.46 cm (short) is the critical stimulus, with both lines being 1.8 cm (long) as the non-critical stimulus. In the second condition, these are reversed. The third and fourth conditions are simultaneous-discrimination tasks, meaning the participant is provided both the critical and non-critical stimulus at the same time for comparison. In the third condition, the critical condition occurs when the lines are different in length while in the fourth condition, these are reversed. Critical stimuli are sequenced such that there are at least four non-critical stimuli in between each pair of critical stimuli. Each participant completed both simultaneous and successive discrimination conditions (counterbalanced across sessions). Stimulus appeared on screen for 150 ms and total trial duration was randomized to be between 1.3 s and 1.7 s. Figure 3 shows an example of the line stimulus.

### 3.3. Preprocessing and Epoching of EEG Signals

Preprocessing of EEG data was performed through script batch processing using EEGLAB [54] and consisted of a combination of best practice steps from both Makoto’s preprocessing pipeline [55] and the PREP pipeline [56]. Details for these steps can be found in Appendix A but worth noting is that the data is downsampled to 250 Hz, and that EOG is used for Independent Component Analysis (ICA) to remove eyeblink artifacts from the EEG. All tasks were relatively similar in trial duration, ranging from 1 s to 1.7 s, with inter-trial duration ranging from 1.2 s to 2 s. To avoid an epoching window which extends into the following trial for some tasks but not others, a 1 s epoching window was selected based on both trial duration and inter-trial duration. Additionally, analysis performed by the 711 HPW demonstrated that a 1 s window following stimulus-onset contained the majority of EEG activity for each task [18,19]. This resulted in a sequence length of 250 for observations across all three tasks.

For labeling of the EEG signals, trials are divided over the duration of the task into four time segments (bins) for each task, and the BIS measure (described in Section 2.1) is used to determine participant performance for each bin, with BIS values and the corresponding *z*-scores calculated separately for each individual. Using this method resulted in a BIS measure of a participant’s performance for the first, second, third, and fourth quarters of the task, allowing analysis of a participant’s performance as the task progressed in time. Performance for each task and each participant are plotted in Figure 4, including the best-fit line for each task and each participant. From the best fit lines in Figure 4, it can be seen that every participant, for every task, was at their highest performing state in the 1st bin, meaning every bin following the 1st bin was a vigilance decrement in comparison to the 1st bin. However, across the tasks, participants had varying performance following the 1st bin as can be seen in Figure 4, with some experiencing their largest decrement in the 2nd, 3rd, or 4th bins. This makes labelling across all four bins difficult while trying to also maintain a balanced dataset. Given this challenge, we opted to use the 1st and 4th bins for our model creation, labelling the 1st bin as attentive, and the 4th bin as a decrement, resulting in a perfectly balanced dataset.

Proper labeling of the data is crucial for a machine learning model, and utilizing only the 1st bin as attentive maximizes tying the most attentive trials to their respective neural correlates. Additionally, the underlying mechanism that allows success in building a task-generic model is that mental fatigue is consistent in producing a vigilance decrement in these tasks and that it is consistent in its neural correlates across different types of vigilance tasks [16]. As mental fatigue has been shown to accumulate over the duration of a vigilance task, the EEG data for the last bin is most likely to have the neural correlates of mental fatigue. As the last bin is a vigilance decrement for all participants across all tasks, using the 1st and 4th bins should maximize the likelihood that the data is labeled properly and will contain the underlying neural correlates to best ensure its success.

### 3.4. Model Creation

To be effective in detection across participants, a model must be highly generalizable and resistant to the effects of non-stationarity and individual differences. For training and testing of a cross-participant model, this requires that data from participants used for model training must not be used for model validation or testing [20]. This is due to the individual differences and non-stationarity that are inherent within EEG data. If this rule is not followed, the model will likely have overestimated test accuracies, and additionally, the model will not train to be generalizable to a more general population, as the model will learn parameters which are likely only accurate for those participants. Additionally, as this is a task-generic model, the model should be tested with a vigilance task that is unseen by the model. To follow these guidelines, we adopted a leave-two-participants-out cross-validation (L2PO-CV) training method for all three models, resulting in 7-folds. The ATC and line tasks were used to train the model, with the 3-Stimulus Oddball Task used for validation. This L2PO-CV method was used for training and validation of all three models. Both the ATC task and the line tasks have the greatest amount of trials, with each participant having performed four times more trials in each of those tasks than the 3-Stimulus Oddball task, resulting in a more desirable ratio of training to validation data than if the ATC or line tasks were used for validation. Additionally, this ensures there is training data from both experiments to allow additional generalization for the model, as the line task was performed in a separate experiment, with an independently selected pool of participants. Ideally, CV would be performed across all three tasks; however, this was infeasible due to the immense amount of training time it would require. All together this results in training folds with 41 participants and 53,600 observations total and validation folds with 2 participants and 360 observations total.

As these cross-participant models are also task-generic, features must be invariant for not only the participants but also the task. For the frequency-domain model, the average power of the five traditional EEG frequency bands for all 64 scalp electrodes were selected as features, resulting in 320 spectral features for each observation, as literature demonstrated that the average power correlates with mental fatigue and is invariant across task, time, and participant [16]. However, an alternative to performing feature extraction by hand is to have the model extract salient features itself. Recently, autoencoders (AEs) have been shown to be more effective than handcrafted features in their ability to compose meaningful latent features from EEG across various classification tasks [57,58,59]. Another recent deep learning innovation is Temporal Convolutional Networks (TCNs), which are a new type of architecture for time-series data. TCNs have the advantage of processing a sequence of any length without having a lengthy memory history, leading to much faster training and convergence when compared to Long Short-Term Memory (LSTM) models [60]. For the time-domain models, a TCN-AE is used for one of the models, and a TCN for the other. In the next two sections, general information on TCNs and AEs is provided, followed by the proposed architectures, hyperparameters, and training and testing parameters for all three models.

### 3.5. Temporal Convolutional Networks

A TCN is a type of convolutional neural network (CNN) for 1D sequence data and was recently developed by Bai et al. [60]. A TCN utilizes dilated convolutions to process a sequence of any length, without having a lengthy memory history. TCNs are typically causal, meaning there is no information leakage from the future to the past; however, they can be non-causal as well. The primary elements of a TCN consist of the dilation factor *d*, the number of filters *n*, and the kernel size *k*. The dilation factor controls how deep the network is, with dilations typically consisting of a list of multiples of two. Figure 5 provides a visual example of a causal TCN and aids in understanding the dilated convolutions on a sequence, with the dilation list in the figure being (1,2,4,8). The kernel size controls the volume of the sequence to be considered within the convolutions, with Figure 5 showing a kernel size of 2. Finally, the filters are similar as they are in a standard CNN and can be thought of as the number of features to extract from the sequence.

These combined elements form a block as in Figure 5, and blocks can be stacked as they are in Figure 6. This increases the receptive field, which is the total length the TCN captures in processing and is a function of the number of TCN blocks, the kernel size, and the final dilation, as shown in Equation (Equation 6). It is common to have a receptive field which matches the input sequence length; however, the receptive field is flexible and can be designed to process any length, which is a primary advantage of TCNs. Other advantages include their ability to be trained faster than LSTMs/Gated Recurrent Unit (GRU) models of similar length, having a longer memory than LSTMs/GRUs when capacity of the networks is equivalent and having similar or better performance than LSTMs/GRUs on a number of sequence related datasets [60,62]
(6)$$R_{field}=K_{size}\cdot N_{blocks}\cdot d_{final}.$$

### 3.6. Autoencoders

An autoencoder (AE) is a type of neural network architecture for unsupervised learning that is primarily used for reproduction of what is input into the network [63]. This is done through the use of two separate networks. One network named the *encoder*
f(x) compresses the input into a lower-dimensional representation called the *code* or the *latent-space*
h=f(x) and another network named the *decoder* reconstructs the input from the code r=g(h). An example of a standard AE architecture can be seen in Figure 7. Because of the nature of the encoder, AEs are useful for dimensionality reduction, are powerful feature detectors, and can also be used for unsupervised pretraining of deep neural networks [64].

In Figure 7, the code h is constrained to have a smaller dimension than the input x. This is called being *undercomplete* and is typical of an AE, as it forces the AE to capture the most salient features of the training data, and thus, the AE does not overfit the training data and copy it perfectly [63].

### 3.7. Frequency-Domain Model

The frequency-domain model was a fully connected MLPNN as can be seen in Figure 8 and utilized spectral features extracted from the 1s epoched EEG signal using complex Morlet wavelet transforms in MATLAB to determine the mean power of the five traditional frequency bands: delta (2–4 Hz), theta (4–7 Hz), alpha (8–12 Hz), beta (13–29 Hz), and gamma (33–80 Hz) (details of this process are out of scope for this paper, and we refer the reader to Chapters 12 and 13 in Mike Cohen’s book, *Analyzing Neural Time Series Data* [66]). With 64 channels from the 64 electrode cap, this resulted in 320 spectral features for each observation (5×64=320). To improve model training, the spectral features were standardized and also log transformed.

The model consisted of three hidden layers with hidden units hu, each followed by a dropout layer with dropout rate dr, with the ReLU activation function used for each hidden layer. As specified at the beginning of Section 3.4, L2PO-CV was used for training and validation of the MLPNN model. The Adam optimizer [67] was used to train the models for 300 epochs by minimizing the binary cross-entropy loss, and a hyperparameter sweep was performed over the hidden units hu, the dropout rate dr, and the learning rate lr.

### 3.8. Time-Domain Models

#### 3.8.1. TCN-AE

The TCN-AE architecture was modeled after work done by Thill et al., who recently developed one of the first published TCN-AE architectures for unsupervised anomaly detection in time series data for health monitoring of machines [68]. They credit the success of this model architecture to the architecture’s ability to compose and encode salient latent features from the data, doing so unsupervised. This architecture involves first training the AE to have the ability to reconstruct the EEG signal with minimal loss. Then the encoder of the trained AE encodes the EEG signal to its latent representation, and those latent features are used for training of a classification model. Their architecture was used as a basis for the TCN-AE model of this research, as the goal for this TCN-AE was to encode the most salient features of the EEG data, and then use those features as input to a fully connected neural network (FCN) classifier to perform classification.

The architecture of the TCN-AE is included below in Figure 9, with the encoder on the left, the decoder on the right, and the latent space in the bottom center. The encoder takes as input the EEG signal with dimensions of 250 × 64, with the 250 representing the sequence length of the 1s epoch downsampled to 250 Hz and the 64 representing the different features from the 64 electrodes. The first layer is a TCN with hyperparameters as specified in Section 3.5, with *d* representing the dilation factor, *k* the kernel size, *b* the number of blocks, and *n* the number of filters. The TCN also used batch normalization, dropout, and recurrent dropout, with the dropout rate dr set as a hyperparameter. This is followed by a 1D convolution (Conv1D) with a kernel size of 1 for further dimensionality reduction and additional non-linearity [68], with *L* representing the number of filters for this convolution layer, which also represents the number of latent features, as there is no further dimensionality reduction after this layer. The ReLU activation function is used for both the TCN and Conv1D layers. Temporal average pooling is then performed with a size of 5 to reduce the sequence length by a factor of 5. This results in the latent space having a sequence length of 50 × *L* number of features.

The decoder is similar to the encoder in its architecture, albeit in reverse. The sequence is first upsampled back to its original length of 250 using nearest neighbor interpolation. The sequence is then passed into a TCN which again has hyperparameters *d*, *k*, *b*, and *n*, followed by a Conv1D layer which increases the dimensionality of the sequence back to its original size of 64. There is no activation function for the TCN and Conv1D layers in the decoder, as this allows the values of the sequence length to take on any value to recreate the original signal.

L2PO-CV was used for training and validation of the reconstruction phase of the AE, with EEG signals standardized by channel for faster model convergence. The Adam optimizer [67] was used to train the autoencoder for 50 epochs for reconstruction of the EEG signal by minimizing the MSE loss, and hyperparameters were grid-searched using Ray Tune version 1.3.0, with the hyperparameters consisting of the dilation factor *d*, the kernel size *k*, the number of blocks *b*, the number of filters *n*, the number of latent features *L*, the dropout rate dr, and the learning rate lr.

Once the autoencoder was trained for reconstruction, the weights of the encoder were locked and the encoder was then used to encode input sequences into latent features. The latent features were then flattened and used as input features into a FCN classifier. The TCN-AE architecture in its entirety can be seen in Figure 10. The FCN classifier had two hidden layers, each with the ReLU activation function, followed by a dropout layer, and a output layer using the sigmoid function. L2PO-CV was used for training and validation of the FCN for classification. The Adam optimizer [67] was used to train the models by minimizing the binary cross-entropy loss, and a hyperparameter sweep was performed over the number of hidden units for each layer, the dropout rate, and the learning rate.

#### 3.8.2. TCN

The TCN model can be seen in Figure 11 and was similar to the encoder portion of the TCN-AE architecture in that it consists of a TCN layer and a Conv1D layer; however, this model differs in that prediction is performed after the Conv1D layer, using an output layer with a sigmoid activation function. The TCN layer has hyperparameters as specified in Section 3.5, with *d* representing the dilation factor, *k* the kernel size, *b* the number of blocks, and *n* the number of filters. The TCN also used batch normalization, dropout, and recurrent dropout, with the dropout rate dr set as a hyperparameter. The Conv1D has a kernel size of 1 and a filter size of 4, providing dimensionality reduction before the output layer. The ReLU activation function is used for both the TCN layer and the Conv1D layer. L2PO-CV was used for training and validation of the TCN for classification, with EEG signals standardized by channel for faster model convergence. The Adam optimizer [67] was used to train the models for 100 epochs by minimizing the binary cross-entropy loss, and a hyperparameter sweep was performed using Ray Tune and grid search over the dilation factor *d*, the kernel size *k*, the number of blocks *b*, the number of filters *n*, the dropout rate dr, and the learning rate lr.

## 4. Results

Below are the results for both the frequency-domain model and the time-domain models. For each model, the best hyperparameter configuration is presented along with its CV balanced accuracy and confidence interval (CI). As accuracy is a binomial distribution, approximate binomial confidence intervals are used. Specifically we utilize Agresti Coull confidence intervals, as they typically maintain α while not being overly conservative [69]. Each model’s CV balanced accuracy and its 95% Agresti Coull confidence interval are compared to random chance, i.e., a naïve classifier with accuracy of 50% (accuracy is 50% as this is a binary classification task). Validation accuracies are also provided for each participant by the participant’s ID, along with their 95% confidence interval. At the end of this section, a table is provided with the participant validation accuracies for each model and the 7-fold CV accuracy for each model.

### 4.1. Frequency-Domain Model

Hyperparameter sweeps for the MLPNN model resulted in the best network achieving a 7-fold CV balanced accuracy of 64% (95% CI: 0.59, 0.69) and 7-fold CV area under the receiver operating characteristic (AUROC) of 0.71 with the following hyperparameters: hidden units of (250, 200, 150) (by layer), learning rate of 0.00001, and dropout rate of 0.5. This results in the model having CV accuracy statistically greater than random chance as evidenced by the confidence interval. Figure 12 depicts the validation accuracies for each participant for the MLPNN model, with nine participants having validation accuracies statistically greater than random chance. Participants 2, 3, 7, 8, and 11 did not have validation accuracies greater than random chance.

### 4.2. Time-Domain Model—TCN-AE

The best hyperparameters found for the TCN-AE signal reconstruction had the following configuration: dilations (1, 2, 4, 8, 16, 32), kernel size of 2, number of filters 36, number of blocks 2, learning rate of 0.0001, and dropout rate of 0.0; and resulted in a receptive field of 2·2·32=128. For the classifier portion of the TCN-AE, all hyperparameter sweeps resulted in similar performance, with accuracies ranging between 48% and 52% for 7-fold CV balanced accuracy, with no set of hyperparameters resulting in a model which performed statistically better than chance. Individual participant accuracies were also investigated for each hyperparameter sweep, with two or less participants having significant performance for the hyperparameter sweeps. No participants had validation accuracies statistically greater than random chance.

### 4.3. Time-Domain Model—TCN

The best hyperparameter sweep for the TCN model yielded a 7-fold CV balanced accuracy of 56% (95% CI: 0.51, 0.61) and 7-fold CV AUROC of 0.57 with the following hyperparameters: dilations (1, 2, 4, 8, 16, 32), kernel size of 4, number of filters 10, number of blocks 2, learning rate of 0.0001, and dropout rate of 0.5; and resulted in a receptive field of 4·2·32=256. This results in the model having CV accuracy statistically greater than random chance as evidenced by the confidence interval. Figure 13 depicts the validation accuracies for each participant for the TCN model, with 3 participants (1, 7, and 12) having validation accuracies statistically greater than random chance.

Table 1 provides the participant validation accuracies and the 7-fold CV accuracy for all three models.

## 5. Discussion

The frequency-domain model (MLPNN) had the highest level of performance of the three model types, with 7-fold CV accuracy significantly greater than random chance at 64% (95% CI: 0.59, 0.69), and nine of the fourteen participants having validation accuracies significantly greater than random chance, as evidenced by their respective 95% confidence intervals. The best time-series domain model (TCN) also had 7-fold CV accuracy significantly greater than random chance at 56% (95% CI: 0.51, 0.61); however, only three of the fourteen participants had validation accuracies significantly greater than random chance. Additionally, the MLPNN had significantly greater CV model accuracy than the TCN model, as evidenced by the 95% confidence interval for the difference between the two classifiers not containing 0, i.e., model accuracy difference of 8% (95% CI: 0.01, 0.15), and the MLPNN also had significantly more participants with validation accuracies greater than random chance than the TCN model, as evidenced by the McNemar’s test statistic of 4.5≥3.84
(p<0.034, α=0.05). Two of the participants in the MLPNN model, 4 and 12, had validation accuracies greater than 80%. Participant IDs of this model that did not have validation accuracies significantly greater than random chance were participants 2, 3, 7, 8, and 11, with Participant 7 having the worst validation accuracy of 46%. Participant 7, however, was the participant with the highest validation accuracy for the TCN model, with Participant 12 being the second highest. Participants having such differing levels of performance across all three model types suggests that low model performance for the TCN and the TCN-AE was not due to certain individual participants having poor quality of data.

One reason for the significant difference between the MLPNN and TCN models could lie in their difference of domains, i.e., frequency vs. time. The literature suggests that changes in the average power of specific bands correlates to mental fatigue in sustained attention tasks [16], which also correlates to a vigilance decrement, and if these spectral features are the most salient information for mental fatigue, then there is no additional information gained by the network utilizing raw time-series signals versus spectral features. Furthermore, TCN performance is contingent on being able to learn that these spectral features are important given only the time-series signals, whereas these spectral features *are* the input for the MLPNN, so the MLPNN does not have to learn them. Thus, the MLPNN may have an advantage over the time-series domain models in that it could already have the most salient features to perform classification.

BIS measures of the 3-stimulus oddball task were investigated to determine if they correlated to model performance of the MLPNN model. If BIS measures are correlated to model performance, this would suggest that the magnitude of decline in a participant’s task performance is correlated to how well the model can classify the EEG; i.e., the worse a decline in a participant’s performance, the better the model can classify the EEG. Additionally, if the MLPNN model uses neural correlates of mental fatigue to perform classification, this would also suggest that as a participant becomes more mentally fatigued, they suffer a larger decline in task performance. To investigate if there was a correlation, BIS slopes of each participant, as well as the difference between the BIS measure of the first and last bins of each participant, were compared to the MLPNN model performance for that participant. These values are provided in Table 2. The BIS slopes and MLPNN validation accuracies were not found to be correlated (ρ=0.07, *p* = 0.82) nor were the BIS difference values and MLPNN validation accuracies (ρ=−0.09, *p* = 0.76).

For the MLPNN model, as this is an artificial neural network, there is no way to know for certain if the model is utilizing neural correlates of mental fatigue to determine if there is a vigilance decrement. However, if the model is utilizing neural correlates of mental fatigue, the lack of correlation between BIS measures and model performance suggests that the magnitude of the mental fatigue does not correlate to the magnitude of the vigilance decrement or that the correlation is participant specific, i.e., some participants could be heavily fatigued and only suffer a slight decrease in performance, while some participants may have a significant decrease in performance when even moderately fatigued. In addition, the vigilance decrement is a measure of task performance, and thus, in general, factors other than fatigue can affect a person’s performance, such as outside distractions, lack of motivation to perform well, etc. It is possible that, even in a lab environment, factors such as this affected participant performance, resulting in a large BIS slope or BIS difference for certain participants, yet with only minimal mental fatigue accumulation.

The literature also notes that the neural correlates of mental fatigue and sleep fatigue manifest differently depending on the task and that they can be opposites of one another, yet both types of fatigue affect task performance in a similar manner. Given this, it could be that some of the participants accumulated sleep fatigue, as opposed to mental fatigue, as the task continued on, resulting in a decrease in performance but with neural correlates which differ from mental fatigue. As these neural correlates can be opposites of one another (e.g., an increase in spectral power for the alpha band as opposed to a decrease), it would be difficult for the model to generalize both of these types of fatigue.

Additional challenges associated with this work include that each of the vigilance tasks used in the datasets were visual tasks as opposed to other types of tasks (e.g., auditory). In order to further validate the model, data from vigilance tasks outside of the visual domain should be used to test the model. Additionally, in order to build a truly task-generic model, training data will likely be needed from these different task domains. To continue to properly validate the model with an unseen task, this would require at least two tasks worth of data from each of the different task domains (one for training and one for validation), requiring a diverse amount of data from many different experiments.

## 6. Conclusions and Future Work

In conclusion, the model type that was most capable of classifying the vigilance decrement in an unseen task and unseen participant out of the models examined was the MLPNN frequency-domain model, utilizing spectral features extracted from the EEG, namely the average power of the five traditional EEG frequency bands. This finding is significant as thus far, a task-generic EEG cross-participant model of the vigilance decrement, i.e., a model capable of classifying the vigilance decrement in an unseen task and unseen participants, has not been built or validated. Previous work by Yin et al. in building a task-generic model did not utilize a different type of task in order to validate their model and instead only varied parameters within a single task of operating an aircraft’s cabin air management system in order to create two tasks. In contrast, the advantage with our research is that it utilized three different types of tasks (the air traffic controller task, the line task, and the 3-stimulus oddball task), all of which are well established in the literature as vigilance type tasks. Additionally, having utilized two tasks for training from two separate experiments as opposed to only one task is likely to provide additional generalization of the model.

To improve model performance, future work should incorporate more vigilance tasks for both training and testing as more EEG vigilance type datasets become available. CV should also be performed across all tasks to investigate if certain tasks provide more or less generalization and task invariance to the model.

Selection of specific spectral features, such as certain frequency bands, should also be explored. By selecting only certain frequency bands, and/or certain regions of the head, model performance could be improved, as currently, the model utilizes a large number of features (320), but only certain features or regions of the brain may be needed in order for the model to accurately classify the vigilance decrement, and removing these unnecessary features could reduce overfitting of the model. This feature importance of the neural network model could be determined through visualization techniques which allow for visual inspection of the model features which result in maximum discrimination between the two classes (vigilance decrement vs. not). Further investigation into mental fatigue vs. sleep fatigue could also be useful. Experiments which note the sleepiness of participants throughout the experiment, either through objective measurements such as prolonged eye closure, or through subjective measurements such as observation and surveys, could result in separate data for neural correlates of mental fatigue vs. sleep fatigue. These experiments could then be used for separate training and testing of the model, and this could reveal if incorporating both types of fatigue either aids the model or hinders it.

To further validate the model, future experiments should investigate devising tasks which result in an increase in vigilance. Currently, every participant experiences a vigilance decrement over the duration of each task, as the tasks are designed to do so. However, this presents a concern for model validation as the data is homogeneous across every participant. Ideally, for model training and validation, there would be data for both a vigilance decrement and a vigilance increase to ensure the model could differentiate between the two and to ensure the model is not classifying based solely on task duration. These tasks could perhaps be achieved through planned breaks throughout the task; however, these experiments would require further validation themselves to ensure they reliably produce an increase in vigilance.

Separate but related work which should stem from this research would be to use EEG to determine when an individual is dropping below a standard level of performance. The vigilance decrement is useful as it informs when someone is experiencing a decrease in performance; however, this decrease in performance is relative to the person’s own baseline level of performance. In certain tasks, it would be valuable to predict when an individual’s expected performance would be too low for successful task completion. This research has demonstrated that EEG can be utilized to determine whether or not someone is experiencing a vigilance decrement, even in an unseen task, and thus it is possible that a model could utilize a participant’s baseline measure of performance to determine if that participant has dropped below a performance threshold; however, more work is necessary for proper implementation. Additionally, a regression model could be investigated to predict the measure of performance itself.

Lastly, EEG research into the vigilance decrement should overall move towards more multi-task experiments, task agnostic models, and dataset sharing. This research demonstrated that the neural correlates of the vigilance decrement span across different task types and can be utilized to detect the vigilance decrement across these task types; however, to further pinpoint which specific features span across all of the different types of vigilance tasks, additional experiments which utilize multiple tasks are needed. Dataset sharing through repositories such as Kaggle [70] or the UCI machine learning data repository [71] would also further enable future research into task agnostic models, as experiments with different tasks could be combined for model building and neural correlate analysis. Future experiments require time and funding; however, dataset sharing could quickly enable this research by utilizing existing datasets across many types of vigilance tasks (piloting of aircrafts, driving, air traffic control, etc.).

## Figures and Tables

**Figure 1 sensors-21-05617-f001:**
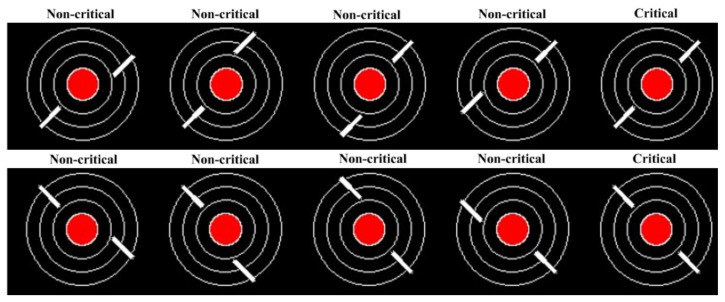
Examples of the different Air Traffic Controller Task stimuli [19].

**Figure 2 sensors-21-05617-f002:**
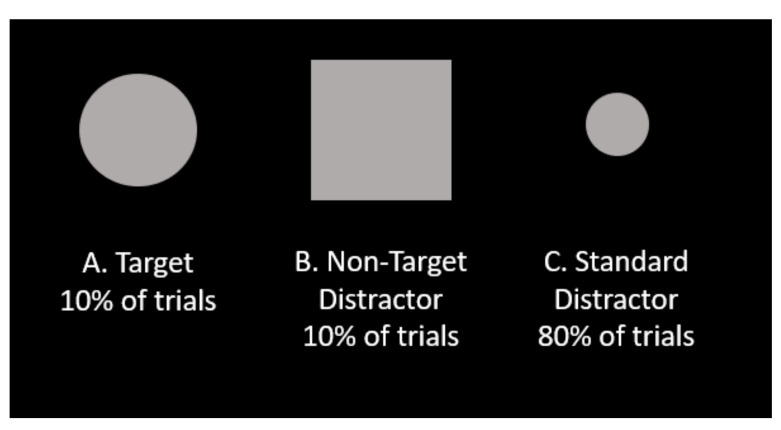
Shapes for the 3-Stimulus Task. The target is a large circle (**A**), the non-target distractor a large square (**B**), and the standard distractor a small circle (**C**) [18].

**Figure 3 sensors-21-05617-f003:**
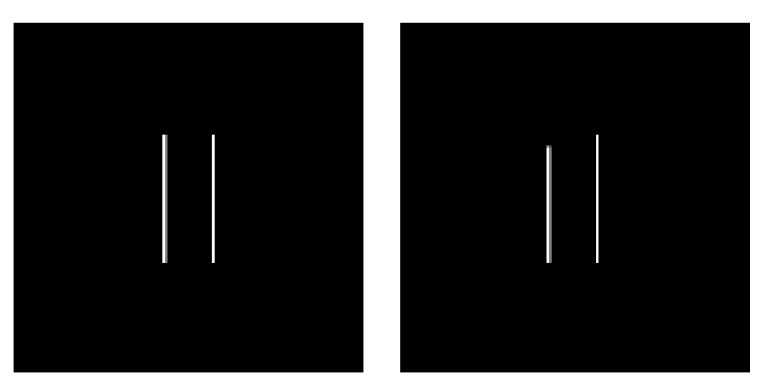
Examples of the different line task stimuli, with lines being the same length on the left and different lengths on the right.

**Figure 4 sensors-21-05617-f004:**
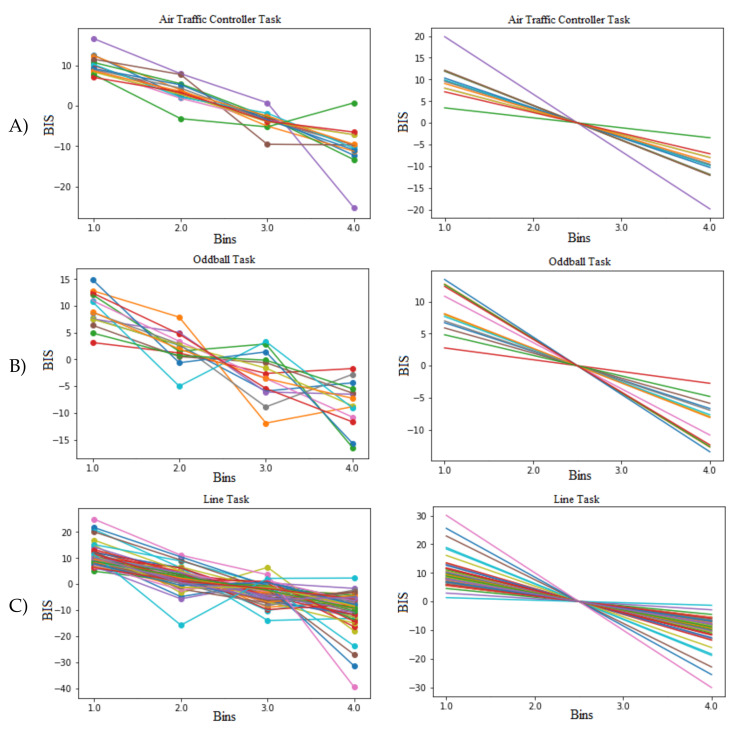
BIS measures and the corresponding best-fit lines for: (**A**) Air Traffic Controller task (top), (**B**) Oddball task (middle), and (**C**) Line task (bottom). The lines represent a participant’s BIS measures over the duration of the task, with lines on the right being best-fit lines. BIS measures vary from bin to bin for each participant, with some participants decreasing steadily throughout the entire task, some decreasing initially and then recovering, or some alternating between decreasing and increasing BIS. Note that every participant’s best-fit line has a negative slope, indicating that every participant’s first bin is their most attentive bin with their largest BIS measure.

**Figure 5 sensors-21-05617-f005:**
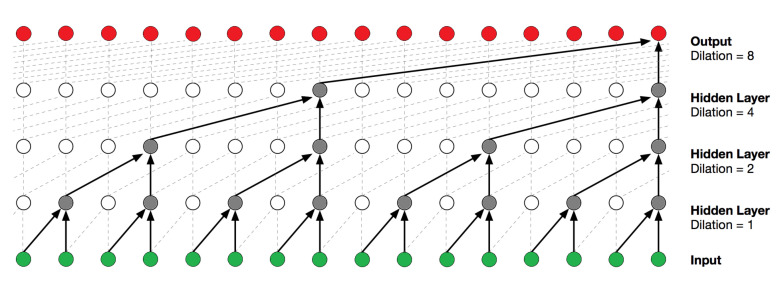
Visual illustration of a causal TCN [61]. This TCN has a block size of 1, a dilation list (1,2,4,8) (i.e., dilation factor 8), and a kernel size of 2. This results in a receptive field of 2·1·8=16.

**Figure 6 sensors-21-05617-f006:**
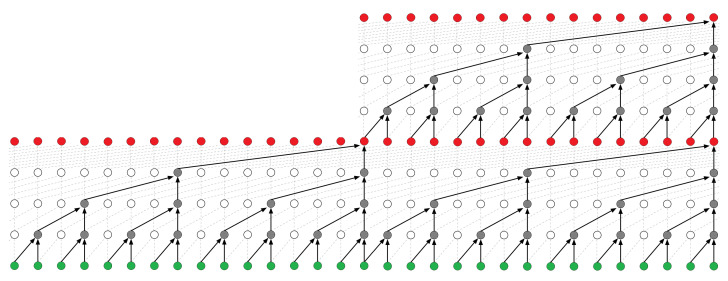
Visual illustration of a causal TCN with stacked blocks [62]. This TCN has a block size of 2, a dilation list (1,2,4,8) (i.e., dilation factor 8), and a kernel size of 2. This results in a receptive field of 2·2·8=32.

**Figure 7 sensors-21-05617-f007:**
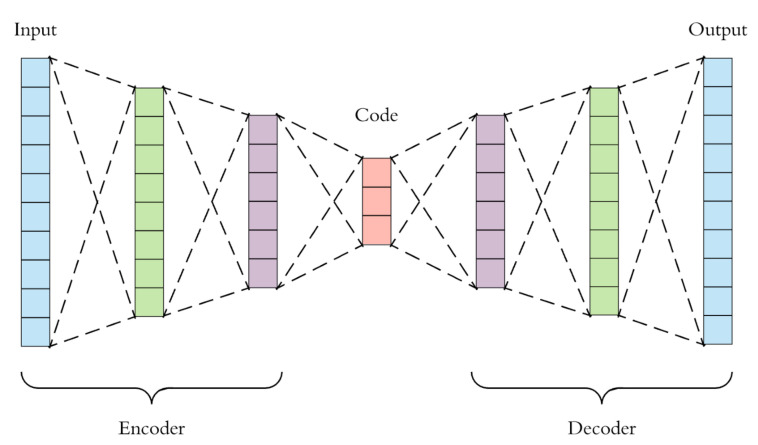
Visual representation of a standard AE architecture [65].

**Figure 8 sensors-21-05617-f008:**
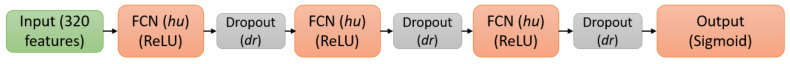
Visual representation of MLPNN classifier. The MLPNN architecture consists of three fully-connected hidden layers with hidden units hu and the ReLU activation, each followed by a dropout layer with a dropout rate dr.

**Figure 9 sensors-21-05617-f009:**
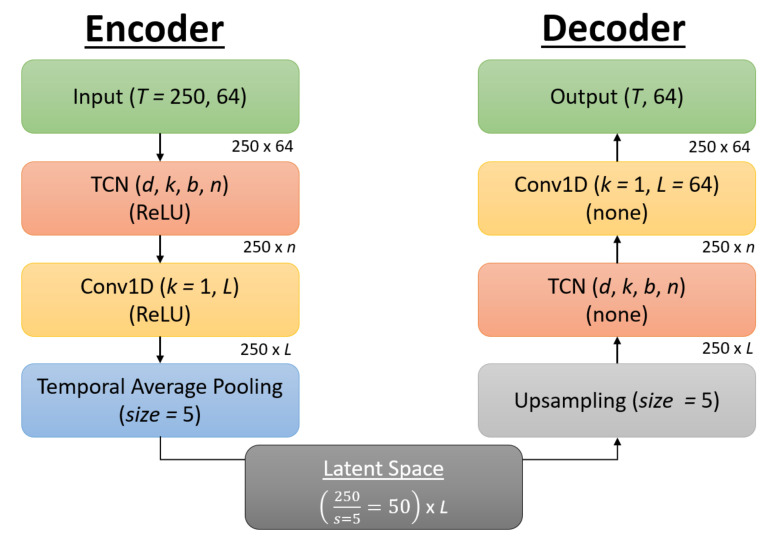
Visual representation of the TCN-AE architecture. Each block corresponds to a layer, with hyperparameters for that layer *italicized*. The activation function for the TCN and Conv1D layers is in parentheses, using ReLU for the encoder and no activation function for the decoder. The dimensions for the input are also provided in the upper-right of each layer as it passes throughout the architecture, with the dimensions starting at *T* = 250 for the sequence length and 64 representing the features (corresponding to the 64 electrodes). The latent space dimensions are 50 × *L*, with *L* being a hyperparameter.

**Figure 10 sensors-21-05617-f010:**
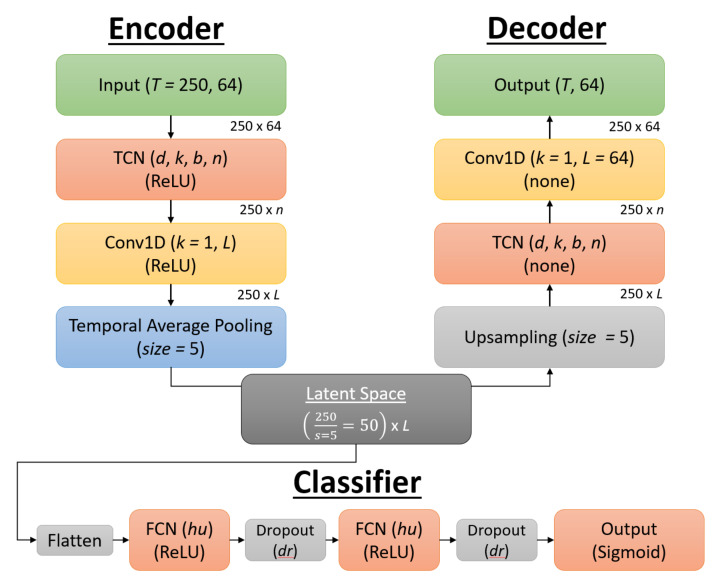
Visual representation of the TCN-AE classifier. The Encoder and Decoder comprise the AE architecture, with the latent space then used as input to the FCN classifier shown at the bottom. The FCN classifier architecture consists of two fully-connected hidden layers with hidden units hu, each followed by a dropout layer with a dropout rate dr.

**Figure 11 sensors-21-05617-f011:**
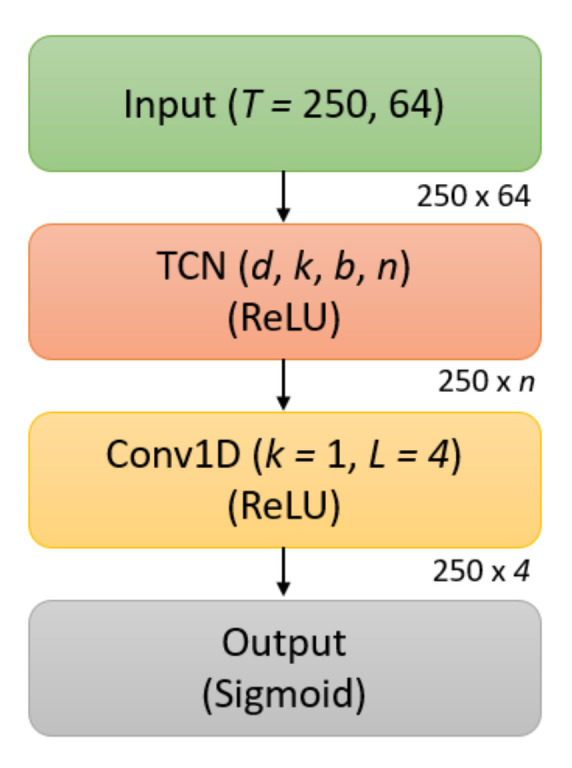
Visual representation of the TCN classifier. Each block corresponds to a layer, with hyperparameters for that layer *italicized*, and the activation function in parentheses.

**Figure 12 sensors-21-05617-f012:**
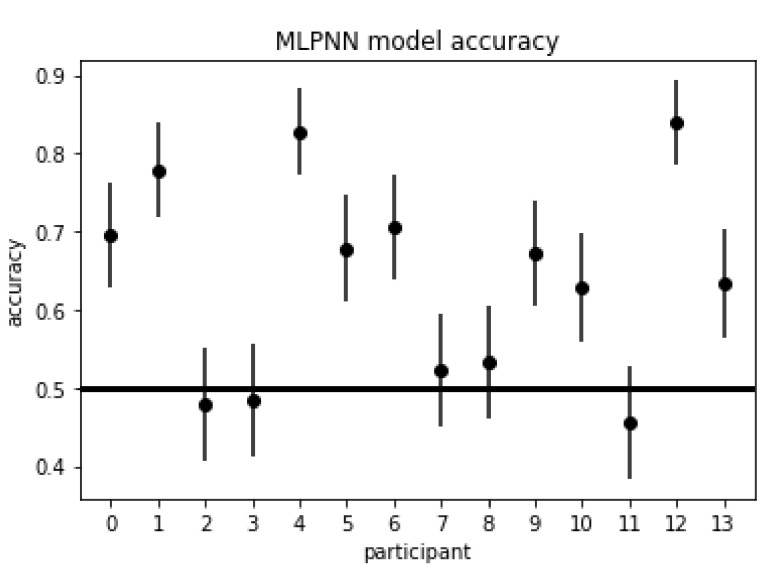
Participant validation accuracies for the MLPNN model, with 9 participants having validation accuracies statistically greater than random chance. Participants 2, 3, 7, 8, and 11 did not have validation accuracies greater than random chance. This model achieved a 7-fold CV accuracy of 64% (95% CI: 0.59, 0.69).

**Figure 13 sensors-21-05617-f013:**
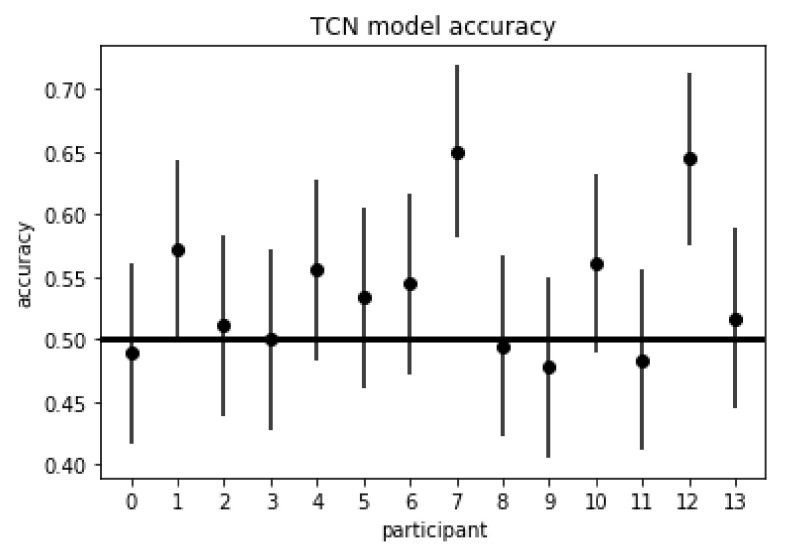
Participant validation accuracies for the TCN model, with 3 participants (1, 7, and 12) having validation accuracies statistically greater than random chance. This model achieved a 7-fold CV accuracy of 56% (95% CI: 0.51, 0.61).

**Table 1 sensors-21-05617-t001:** Vigilance decrement classification model performance results for each model type. Participant validation accuracies and the 7-fold CV accuracy are provided for each model, with 95% confidence intervals provided in parentheses. For both participant accuracies and the 7-fold CV results across all participants, **Bold** signifies statistical significance of accuracy over random chance (defined as 50% for this binary classification task) as evidenced by the 95% confidence interval.

Participant #	MLPNN Val Acc	TCN-AE Val Acc	TCN Val Acc
0	**0.69 (0.62, 0.76)**	0.51 (0.44, 0.58)	0.49 (0.42, 0.56)
1	**0.78 (0.71, 0.83)**	0.49 (0.42, 0.56)	**0.57 (0.50, 0.64)**
2	0.48 (0.41, 0.55)	0.49 (0.42, 0.57)	0.51 (0.44, 0.58)
3	0.48 (0.41, 0.56)	0.52 (0.45, 0.59)	0.50 (0.43, 0.57)
4	**0.83 (0.77, 0.88)**	0.49 (0.42, 0.57)	0.56 (0.48, 0.63)
5	**0.68 (0.61, 0.74)**	0.55 (0.48, 0.62)	0.53 (0.46, 0.60)
6	**0.71 (0.63, 0.77)**	0.47 (0.40, 0.54)	0.54 (0.47, 0.62)
7	0.52 (0.45, 0.59)	0.56 (0.48, 0.63)	**0.65 (0.58, 0.72)**
8	0.53 (0.46, 0.60)	0.56 (0.49, 0.63)	0.49 (0.42, 0.57)
9	**0.67 (0.60, 0.74)**	0.54 (0.47, 0.62)	0.48 (0.41, 0.55)
10	**0.63 (0.56, 0.69)**	0.49 (0.42, 0.57)	0.56 (0.49, 0.63)
11	0.46 (0.38, 0.53)	0.46 (0.38, 0.53)	0.48 (0.41, 0.56)
12	**0.84 (0.78, 0.89)**	0.48 (0.41, 0.55)	**0.64 (0.57, 0.71)**
13	**0.63 (0.56, 0.70)**	0.53 (0.46, 0.60)	0.52 (0.44, 0.59)
7-fold CV	**0.64 (0.59, 0.69)**	0.52 (0.47, 0.57)	**0.56 (0.51, 0.61)**

**Table 2 sensors-21-05617-t002:** This table provides the BIS slope and difference between the BIS measures of first and last bin for the oddball task for each participant. Validation accuracy for the frequency-domain MLPNN model is also provided for each participant. **Bold** signifies statistical significance of accuracy over random chance (defined as 50% for this binary classification task) as evidenced by the 95% confidence interval.

Participant #	BIS Slope	BIS Difference (1st Bin–4th Bin)	MLPNN Val Acc
0	−4.46	12.02	**0.69 (0.62, 0.76)**
1	−8.48	21.67	**0.78 (0.71, 0.83)**
2	−8.42	28.53	0.48 (0.41, 0.55)
3	−1.84	4.85	0.48 (0.41, 0.56)
4	−5.34	14.07	**0.83 (0.77, 0.88)**
5	−3.92	12.68	**0.68 (0.61, 0.74)**
6	−7.22	21.75	**0.71 (0.63, 0.77)**
7	−4.64	11.56	0.52 (0.45, 0.59)
8	−5.30	16.19	0.53 (0.46, 0.60)
9	−5.11	19.81	**0.67 (0.60, 0.74)**
10	−8.96	30.53	**0.63 (0.56, 0.69)**
11	−5.40	16.16	0.46 (0.38, 0.53)
12	−3.21	10.41	**0.84 (0.78, 0.89)**
13	−8.24	24.08	**0.63 (0.56, 0.70)**

## Data Availability

This work uses only de-identified existing data. A Human Research Protection Plan review determined this research to be not involving “human subjects” under US Common Rule (32 CFR 219) on 6 Jun 2020. The EEG datasets are owned by the United States Air Force Research Laboratory, 711th Human Performance Wing (HPW) and were collected through experiments performed by the 711th HPW in partnership with the University of Dayton. See [18,19] for details surrounding the experiment which produced the ATC and 3-stimulus Oddball Task dataset.

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
