# Peer review of "Generalized Deep Learning EEG Models for Cross-Participant and Cross-Task Detection of the Vigilance Decrement in Sustained Attention Tasks"

_sensors, 2021, doi:10.3390/s21165617_

Round 1

Reviewer 1 Report

This paper explored the creation and application of a task-generic EEG cross-participant model for detection of the vigilance decrement in an unseen task and unseen individuals. The manuscript is well written and has a significant contribution to the field of study. However, the following comments should be addressed to improve the presentation of the manuscript.

  1. The introduction and related work are very long and very descriptive. These sections should be reduced and focused more in recent studies on vigilance decrement.
  2. For section 2.1. Vigilance Decrement. I suggest removing the second and third paragraphs and summarizing them in three sentences while citing recent review work on vigilance decrement.
  3. In Section 2.1.1. Performance Measurement should be reduced to one paragraph.
  4. In Section 2.2. EEG. The literature is not up to date. Authors should add recent studies on vigilance. Many recent papers utilized EEG for vigilance assessment including the following papers:

Al-Shargie, F.M., Hassanin, O., Tariq, U. and Al-Nashash, H., 2020. EEG-based semantic vigilance level classification using directed connectivity patterns and graph theory analysis. IEEE Access8, pp.115941-115956.

Al-Shargie, F., Tariq, U., Hassanin, O., Mir, H., Babiloni, F. and Al-Nashash, H., 2019. Brain connectivity analysis under semantic vigilance and enhanced mental states. Brain sciences9(12), p.363.

  1. In Figure 4. The lines should be labelled within the caption.
  2. The bins (1-4) should be further clarified.
  3. Appendix A could be removed since the preprocessing followed a well-known pipeline.
  4. Some old references should be removed.

Reviewer 2 Report

In this manuscript, the authors present a method for detection of the vigilance decrement in an unseen task and unseen individuals using EEG data. Three different neural network models were proposed and compared. The results have demonstrated possibility to detect a vigilance decrement.

1)         Limitation: all tasks are based on the analysis of visual information. This restricts the generalizability of the model to other types of tasks (memory, audio, sensory, motor, thinking, etc.) which also could cause a vigilance decrement.

2)         No interpretation of the models is provided. It would be very interesting to make analysis of the prediction model to understand what factors (electrodes, bands) mostly influence the decision. Interpretation of the neural network model is not easy, but some discussion would be useful.

3)         The informativeness of the frequency-domain features can be increased and made less subject dependent if these features would be normalized on the corresponding baseline resting features:

feature_norm_i = (feature_i – feature_baseline_i) / feature_baseline_i

Otherwise, the energy in different frequency bands could significantly depend on subjects, electrode resistance, etc.

4)         Random chance should be evaluated using the permutation test (random shuffling of the labels in the training set, and then you do the same procedure as with unshuffled data). Further, statistical test (e.g., ANOVA) should be applied to detect if your method significantly outperforms the random level.

5)         The prediction quality of the different models in different conditions (between subjects, between tasks) should be compared using statistical tests. In general, statistical analysis of the results should be provided.

6) line 550: ‘accuracy statistically greater’ do the authors mean ‘significantly greater’?

Reviewer 3 Report

Dear Authors,

   Many thanks for your manuscript submission. After my review, I think this paper comprehensively established a wonderful set of research work. The key idea is to exploit three different dep learning model for task-generic EEG cross-participant model when detecting the vigilance decrement in unseen tasks / individuals. If everything presented in this article is original, a few other aspects are recommended for minor edits: 

   a) Reduce the number of keywords; 4~6 is good enough, 6 is maximum.

   b) Add a short paragraph for summarizing the main contribution of your research work in the Introduction section. 

   c) Emphasize any technical innovations of your three deep-learning models (MLPNN, TCN, and TCN-AE), if any; or specify any other original ideas when handling the EEG cross-participant tasks for unseen targets.

   d) Adjust the 1st column of Tables 1-2 with middle-alignment, if applicable.

   e) Re-define some metrics in Subsection 2.1: Performance Measurement, i.e., Accuracy, response time (RT) in formula shape.

   f) Add a short paragraph to distribute your limitations of study (including challenging tasks and opening questions) in the last section (before stating future work), and adjusting some of the paragrah arrangements.

   g) References: sufficient citations are displayed in this version, while the use of abbreviated terms should be posted on journal citations (as well as some highly visible conferences); meanwhile, adding a few more references published in Sensors / Entropy / Remote Sensing / Technology / Appliced Science of MDPI affiliated journal publications in Years 2020-2021 are also recommended. That would increase your possibility of paper acceptance.

   Thanks so much again for your careful edits towards paper acceptance. We expect your success in the near future. Good luck!

Best wishes,

Yours faithfully,

Reviewer 4 Report

This study aimed to utilize deep learning algorithms to classify vigilance decrement levels using EEG.  I have the following suggestions.

  1. Please add a paragraph about the contribution of this article in bullet form at the end part of the Introduction section.
  2. Authors should report ethical or IRB declarations related to data acquisition of this study.
  3. In line 254, the authors described that two reference electrodes were placed over the mastoids. I am curious about ground electrodes. Would you please clarify the positions and numbers of reference and ground electrodes?
  4. Subject demographics need to be reported. The authors should describe the study protocol in detail.
  5. The authors mentioned that they measured EOG also. For this study, the role of EOG should be discussed in the methodology section. You said it in the appendix section, but the purpose is not clear.
  6. Authors should report the statistics of EEG frequency-domain and time-domain features for each Vigilance Decrement Task. Would you please add the errorbar plots of EEG frequency-domain and time-domain features for each Vigilance Decrement Task in the result section? Describing feature statistics is important before implementation in deep learning models.
  7. It is essential to investigate the significant EEG features for Vigilance Decrement Tasks.
  8. It is expected that ML/DL may clasify the cognitive outcome for vigilance decrement. EEG-based ML/DL has been implemented for task-induced neurological outcomes, disease prognostics, brain stimulations and so on. References should be improved by adding related articles, such as doi:10.3390/brainsci11070900. The contribution and the findings of this research should be better compared with the state-of-the-art.
  9. Both training and testing model ROC curves need to be shown. You should add more performance measures of classifiers, such as, sensitivity, specificity, and precision from the confusion matrix.
  10. Authors must discuss the advantages and drawbacks of their proposed method with other studies adding a discussion section.

Round 2

Reviewer 2 Report

No additional comments